# Advancing Antibody Titer Assessment in ABO-Incompatible Transplantation

**DOI:** 10.3390/antib14030078

**Published:** 2025-09-15

**Authors:** Masayuki Tasaki, Kazuhide Saito, Kota Takahashi

**Affiliations:** 1Division of Urology, Department of Regenerative & Transplant Medicine, Graduate School of Medical and Dental Sciences, Niigata University, 1-757, Asahimachi-dori, Chuo-ku, Niigata 951-8510, Japan; kazsaito@med.niigata-u.ac.jp; 2Takahashi Memorial Medical Institute, Tokyo 170-0004, Japan; takahashi2310@outlook.jp

**Keywords:** anti-ABO antibody titer, hemagglutination-based methods, glycan microarray, CD31, ABO-incompatible transplantation

## Abstract

**Background**: The accurate evaluation of anti-ABO antibodies is essential for risk stratification in ABO-incompatible (ABOi) transplantation. Historically, hemagglutination-based titration has been the cornerstone of such an assessment; however, different tools are being evaluated in this context. In recent years, several novel methods have been reported. **Methods**: A narrative review was conducted using PubMed, Scopus, and Google Scholar, focusing on recent studies evaluating anti-ABO antibody measurement techniques in the context of ABOi organ transplantation. **Results**: In addition to the conventional tube method, techniques such as column agglutination technology, flow cytometry, and enzyme-linked immunosorbent assay are utilized for anti-ABO antibody assessment. However, any particular technique, significant interinstitutional and interoperator variabilities have been reported due to differences in the detailed protocols and the inherently subjective nature of some techniques. Moreover, these assays are based on the antibody binding to ABO antigens expressed on red blood cells, which might not accurately reflect the clinical context of organ transplantation. In recent years, technological advances have enabled the development of novel assays evaluating antibody responses specifically against the ABO antigens expressed on vascular endothelial cells. These include glycan microarrays, which differentiate responses by ABO antigen subtypes, and CD31-based microarrays, wherein recombinant CD31 proteins expressing ABO antigens are immobilized. These approaches are applied to assess clinically relevant anti-ABO antibodies in the context of ABOi organ transplantation. **Conclusions**: The objective evaluation of antibody titers against ABO antigens on vascular endothelial cells might not only enable a more accurate risk assessment but also facilitate meaningful comparisons between institutions.

## 1. Introduction

Technological advances in human leukocyte antigen (HLA) laboratory testing have undoubtedly improved the sensitivity and specificity of HLA antibody assessment. Multiple methodologies, such as complement-dependent cytotoxicity test, flow cytometry, and Luminex-based technology, can be available for the HLA antibody test. The understanding of complement (C1q and C3d, etc.) fixing antibodies and IgG subclass in HLA antibodies has become widespread. In contrast, antibody tests against ABO antigens in ABO incompatible (ABOi) organ transplantation are still primitive. The accurate evaluation of anti-ABO antibodies is essential for risk stratification, such as acute antibody-mediated rejection (ABMR) in ABOi organ transplantation. Isohemagglutinin assays employing red blood cells (RBCs) are the most common assay used to measure antibody titer in ABOi organ transplantation. In recent years, several novel methods have been reported. In this review, we summarize the methods for measuring anti-ABO antibodies and discuss their challenges based on previous reports.

## 2. ABO Blood Group System

The 1930 Nobel Prize in Physiology or Medicine was conferred upon Dr. Karl Landsteiner in recognition of his seminal contribution to the understanding of human blood group systems. In his pioneering work, first reported in 1901 [1], Landsteiner conducted a series of in vitro agglutination assays using his own blood and that of his laboratory colleagues. He observed that mixing erythrocytes and plasma from different individuals resulted in reproducible hemagglutination patterns contingent upon the specific combinations utilized.

Based on these observations, Landsteiner classified the agglutination patterns into distinct groups, designating them as “A” and “B”. Certain RBC samples notably exhibited no agglutination with any tested plasma and were initially labeled as group “C”, but their designation was subsequently replaced by “O”. It was possibly derived from the numeral zero and symbolized the absence of reactive antigens. This discovery laid the foundation for the now universally recognized ABO blood group system.

Subsequent investigations conducted within Landsteiner’s laboratory led to the identification of the AB phenotype, which was characterized by RBCs that reacted with most plasma samples and plasma that failed to induce agglutination in other erythrocyte populations [2].

Landsteiner’s Law, which was derived from these early experiments, postulates that individuals naturally produce isohemagglutinins (anti-A and/or anti-B antibodies) against the ABO antigens absent on their own erythrocytes [3]. Accordingly, individuals with blood groups A and B possess anti-B and anti-A antibodies, respectively. Those with blood group O express both anti-A and anti-B antibodies. Finally, those with blood group AB, expressing both A and B antigens, lack any detectable isohemagglutinin.

ABO antibodies naturally occur and differ from the known adaptive immune responses. Unlike immune antibodies that develop as immunoglobulin M (IgM) before switching to immunoglobulin G (IgG) when exposed to antigens, ABO antibodies often persist as IgM. In particular, most type A and type B ABO antibodies are IgM [4]. IgG antibodies are found relatively frequently in type O, with the IgG2 subclass as the main component [5]. The cause of naturally occurring ABO antibodies without exposure to ABO antigens is the cross-reaction with similar antigens present in the environment, especially bacteria. Hence, the ABO antibody titer is well known to be elevated when exposed to bacterial antigens [6].

## 3. ABO-Incompatible Kidney Transplantation (ABOi KTx)

The ABO blood group antigens are expressed in various organs, including the kidneys. Within the kidneys, type A and type B antigens are expressed on the endothelial cells of the arteries, veins, glomerular capillaries, and peritubular capillaries, as well as the epithelial cells of the distal tubules and collecting ducts [7]. Landsteiner’s Law applies similarly in the context of kidney transplantation (KTx). In ABOi KTx, naturally occurring anti-A and/or anti-B antibodies present in the recipient’s circulation can recognize and bind to the corresponding ABO antigens expressed on the renal allograft. This interaction might trigger antibody-mediated rejection (ABMR), consequently contributing to early graft injury and potentially compromising transplant outcomes.

ABOi KTx was first performed by Hume and colleagues in 1952 [8]; however, the renal graft did not function. Starzl and colleagues performed it in two patients in 1964, and achieved long-term graft survival in one B-incompatible KTx patient [9]. Wilbrandt and colleagues reported 12 ABOi KTx cases in 1969, with acute rejection in all patients post-graftectomy [10]. The anti-donor blood group antibodies increased in these patients after ABOi KTx, suggesting that antibodies against the donor blood group antigen were produced by stimulation with the donor ABO antigen expressed in the renal graft, resulting in acute ABMR [10].

Alexandre and colleagues were the first to design a transplant procedure using plasma exchange (for pretransplant antibody removal) combined with splenectomy [11,12,13]. This finding suggested that antibody reduction and suppression before ABOi KTx were critical for a successful transplant. In Japan, Takahashi and colleagues achieved long-term graft survival in ABOi KTx cases by applying pretransplant antibody removal therapy using double-filtration plasmapheresis (DFPP) combined with simultaneous splenectomy during transplantation [14]. In Japan, where organ donation is extremely limited compared to Western countries, ABOi KTx has been widely adopted ahead of the rest of the world as a means to expand transplant opportunities for patients with end-stage renal disease. Takahashi et al. reported long-term outcomes in 441 patients who received ABOi living donor kidney transplants between January 1989 and December 2001 [15]. No significant differences were found between the A- and B-incompatible recipients with respect to clinical outcomes. Thus, they concluded that ensuring long-term outcomes in the recipients of ABOi KTx was an effective treatment for end-stage renal disease.

Since then, ABOi KTx has been adopted in many countries. The key factors contributing to the procedure’s success include the advancement of desensitization protocols and the ability to predict ABMR based on antibody titers.

## 4. Antibody Titer Assessment

ABO antibodies are usually titrated with serial two-fold dilutions of serum with selected RBCs. This approach is widely used in several clinical laboratories because of its long history, presence of practitioners with extensive related experience, and its characteristic of not deviating from the basic framework of other transfusion-related tests. Titration values can provide information on the relative amount of antibodies present in the serum. Accordingly, monitoring titers is important to determine the effectiveness of desensitization before ABOi KTx and predict or diagnose acute ABMR post-ABOi KTx.

However, titration has several inherent limitations. For instance, this approach has not yet been standardized, and significant interobserver and interlaboratory variations exist. Titration techniques vary across laboratories in terms of the types of methods used and whether they measure IgM or IgG. The semiquantitative nature of titration is another clinical disadvantage. Reagent cell selection is another aspect that can influence the standardization of titration. Titration protocols vary depending on the reagent cell, diluent, presence of anti-human globulin or dithiothreitol (DTT), incubation time, and reaction temperature. In addition, no consensus has yet been reached regarding the agglutination strength used to define the endpoint in antibody titer determination, with some reports considering 1+ agglutination as the cutoff while others considering weak positive (± or w+) reactions as the endpoint for the conventional tube method and column agglutination technique (CAT). This lack of standardization might contribute to interlaboratory variability and complicate the comparison of titer results across studies. Thus, comparing the antibody titer results of different laboratories is challenging. In 2008, the College of American Pathologists proposed a uniform procedure to reduce such variations and standardize these methods [16]. In this study, the authors modified the conventional ABO antibody titration technique by implementing a standardized protocol across multiple laboratories, including uniform reagents, incubation times, and interpretation criteria. Consequently, the interlaboratory variation in anti-A titers was significantly reduced, demonstrating that procedural harmonization can improve the reproducibility and reliability of antibody titer measurements. Such detailed refinements and standardization of antibody titration procedures can effectively reduce the inter-institutional variability to a certain extent.

In the context of ABOi transplantation, the accurate determination of IgG anti-A or anti-B antibody titers is critical for assessing the risk of ABMR. However, coexisting IgM antibodies can interfere with IgG-specific titer measurements if they are not properly inactivated. IgM is a pentameric immunoglobulin possessing a strong agglutinating capacity at room temperature. It can directly cause RBC agglutination independent of antiglobulin reagents. In contrast, IgG antibodies have a lower avidity and typically require enhancement at 37 °C, followed by detection with anti-human globulin (Coombs reagent) to induce visible agglutination. If IgM antibodies remain active during IgG titer assays, particularly during incubation or agglutination reading steps, they may induce red cell agglutination independent of IgG, leading to false-positive or overestimated IgG titers. Therefore, selective IgM inactivation, typically using DTT, is an essential step for isolating the IgG-specific response.

Several methods have been employed to inactivate IgM antibodies when measuring anti-A and anti-B antibody titers, particularly in the context of ABOi transplantation, where selective IgG detection is clinically relevant. The major approaches are as follows:**DTT treatment**: DTT disrupts the pentameric structure of IgM by reducing disulfide bonds, thereby effectively abolishing its agglutinating activity while preserving IgG structure and function. It is considered the gold standard for IgM inactivation because of its high reproducibility and minimal impact on IgG.**2-Mercaptoethanol (2-ME)**: Similarly to DTT, 2-ME reduces the disulfide bonds in IgM. However, it is used less frequently in modern laboratories due to its volatility and higher toxicity.**Heat inactivation**: Incubation at 56 °C for 30 min can denature IgM; however, this method might also impact IgG stability and other serum components, such as complement, leading to variable results.**Enzymatic digestion**: Proteolytic enzymes such as pepsin can degrade IgM; however, these enzymes often lack specificity and may also degrade IgG, which limits their utility in clinical practice.**Immunoaffinity-based removal**: Columns conjugated with proteins such as Protein L, G, or A can selectively bind and remove specific immunoglobulin subclasses. While highly specific, these methods are technically demanding, costly, and not routinely applied in clinical laboratories.

Among the abovementioned methods, DTT treatment remains the most widely adopted and reliable technique for IgM inactivation in serological testing. Its efficacy in selectively eliminating IgM activity while preserving IgG functionality makes it the preferred approach for accurate and reproducible antibody titer determination.

In the ABOi KTx setting, IgG antibody titers are reported to be clinically important. Therefore, an accurate assessment of IgG levels is essential. IgM inactivation is a necessary step of this assessment; however, it does not appear to be routinely implemented. In clinical laboratories, it is common practice to measure IgM titers using the saline tube method with incubation at room temperature, while IgG titers are assessed using the indirect antiglobulin test with incubation at 37 °C without IgM inactivation and with anti-IgG reagent. This two-tiered approach allows for the differential evaluation of IgM and IgG antibody activities and is widely adopted, particularly in transplant centers managing ABOi KTx. According to the “uniform procedure” for ABO antibody titration suggested by the College of American Pathologists, the tube test can be converted to the AHG phase after 30 min of incubation at room temperature without DTT treatment [16]. Therefore, if DTT has not been used for the titration of the IgG antibody, the results must be reported as a measure of “total antibodies” and not “IgG” [16,17].

## 5. Conventional Tube Method (CTM)

CTM remains the most widely used and standardized approach for anti-A and anti-B antibody titration in the context of ABOi transplantation. Despite the development of more automated or sensitive platforms, such as gel cards and flow cytometry-based assays, CTM continues to be favored in many clinical laboratories due to its simplicity, low cost, and long-standing clinical utility. Importantly, a substantial amount of clinical outcome data has been accumulated using CTM-based titers, allowing clinicians to interpret results in the context of established thresholds for immunosuppressive therapy, desensitization protocols, and graft survival. This extensive experience provides a practical advantage because many transplant programs have historically defined “safe” pretransplant antibody titer levels based on CTM results. Furthermore, CTM can be performed without specialized equipment, making it accessible even in resource-limited settings. Thus, CTM remains the reference method in several countries, particularly in Japan, where ABOi transplantation is commonly practiced, and national proficiency testing is conducted using tube-based titration. Despite its widespread use and clinical familiarity, CTM has several notable limitations. One of the primary concerns is its reliance on the subjective visual interpretation of the agglutination strength by individual technicians, which can lead to significant interoperator and interlaboratory variabilities. The lack of objective, instrument-based readouts reduces reproducibility, particularly in borderline or weakly positive reactions. Moreover, the technical steps involved in CTM, such as RBC suspension concentration, incubation time and temperature, serum-to-cell ratio, and whether or not IgM is inactivated (e.g., with DTT), often vary across laboratories. This procedural heterogeneity complicates standardization efforts and makes the cross-comparison of titers between institutions unreliable.

Our institutional protocol for CTM is described below. Briefly, recipient serum or plasma is serially diluted with saline solution, and 3% of the RBC suspension (50 μL) is mixed with the diluted plasma sample (100 μL) in a test tube. After agitation using a Vortex mixer, the reaction mixture is allowed to stand for 15–30 min at room temperature (20–25 °C) before the IgM antibody titer is determined. To measure the IgG antibody titer, an equal amount of 0.01 M DTT (Wako Pure Chemical Industries, Osaka, Japan) diluted with 7.4% phosphate-buffered saline (PBS) is incubated with the plasma sample at 37 °C for 30–60 min to inactivate IgM antibody. After washing thrice with saline, the RBCs are incubated with the DTT-treated recipient serum or plasma. After washing thrice with saline again, anti-human IgG antibody (Bio-Rad Anti-human IgG, Bio-Rad Laboratories, CA, USA, for volunteers; Ortho Anti-human IgG, Ortho Diagnostics, Tokyo, Japan, for transplant recipients) is added to the mixture to determine the IgG antibody titer. The mixture is then centrifuged at 900–1000× *g* for 15 s using a Himac centrifuge MC-450 immediately before determining the result. Each method determines the endpoint of agglutination to a 1+ reaction. Although some laboratories use the w+ endpoint for antibody titer assessment, the subjective nature of this reaction strength limits its utility in standardization. We recommend using the 1+ agglutination endpoint for consistency and reproducibility. Figure 1 shows an image of the agglutination reaction obtained using CTM.

In the immediate spin (IS) method, the plasma is mixed with the reagent RBCs and centrifuged promptly. Agglutination is assessed immediately without any preincubation at room temperature. Although this approach confers the advantages of procedural simplicity and rapid turnaround time, it might underestimate anti-A or anti-B IgM antibody titers, particularly in samples with weak or slow-reacting antibodies [16]. IgM-mediated hemagglutination typically occurs at room temperature but may require several minutes to reach visible agglutination. Thus, omitting a 15- to 30 min incubation step at room temperature, as is the case in the IS method, could result in reduced sensitivity for low-affinity IgM antibodies. This limitation has been observed in comparative studies, where the IS method often yielded lower titers than other methods involving room-temperature incubation. Therefore, while the IS method is useful in high-throughput settings, it must be applied cautiously in clinical contexts, such as ABOi transplantation, where an accurate IgM titer quantification is critical for guiding immunosuppressive therapy and risk assessment.

Antibody titer was primarily determined by CTM, that is to say, by visual observation. In this context, subjective rather than objective judgment would most likely cause an interexaminer variation, resulting in an interinstitutional variation. In other words, no common vocabulary or protocols are available for discussion about the anti-A/B antibody titer. The Japanese ABO-Incompatible Transplantation Committee conducted a series of surveys between 2003 and 2004 to assess interinstitutional variations in the measurement of the anti-A/B antibody titer [18]. The interinstitutional difference between the maximum and minimum values reached as much as 32-fold for IgM and 256-fold for IgG. After the introduction of the provisional standard protocol, the intrainstitutional variation was considerably reduced to below eight-fold, except for several institutions that might need further training. Such detailed refinements and standardization of antibody titration procedures can effectively reduce interinstitutional variability to a certain extent, as described in a previous study [16]. However, such efforts must be sustained continuously because laboratory personnel inevitably change over time. In recent years, the Japanese Society for Transplantation and the Japan Society for Histocompatibility and Immunogenetics have jointly conducted an annual external quality assessment for anti-ABO antibody testing, and the results have consistently shown substantial variability in antibody titration among institutions each year.

## 6. Column Agglutination Technique (CAT)

CAT is widely employed in blood group serology analyses, particularly for antibody screening, crossmatching, and isoagglutinin titration in ABOi transplantation. Commercially available CAT systems utilize either a gel matrix or glass beads as the medium within microcolumns. The microcolumn gel and glass bead card tests, based on CAT, were developed in the 1990s. These methods involve centrifuging a column on a reagent card, with agglutinated RBCs becoming trapped in the gel or glass beads and unagglutinated RBCs accumulating at the bottom of the column.

Gel matrix-based CAT (e.g., DiaMed-ID system by Bio-Rad) has become the most commonly used method in many transfusion laboratories due to its high sensitivity, especially for detecting weak antibodies. The gel medium efficiently traps agglutinated RBCs while allowing unagglutinated cells to pass through during centrifugation, thereby providing clear and reproducible results.

In contrast, glass bead-based CAT (e.g., Ortho MTS Gel System) employs a layer of uniformly treated glass particles, presenting a format that offers faster centrifugation and an overall shorter processing time. While some studies suggest that its sensitivity may be slightly lower compared to the gel-based systems, it remains acceptable for routine clinical use and is particularly advantageous in high-throughput settings.

Both systems share the same underlying principle of visualizing agglutination based on RBC retention or passage through the matrix. However, their physical properties and procedural differences can affect performance characteristics, such as sensitivity, specificity, and turnaround time. The selection between gel- and glass bead-based CAT platforms should take into account the laboratory needs, the types of antibodies being evaluated, and the clinical context.

A typical CAT-based protocol is described below. Briefly, recipient serum or plasma is serially diluted with saline solution. For IgM antibody detection, untreated serum is used, whereas for IgG antibody titration, serum or plasma is first treated with a reducing agent, such as DTT or 2-ME, to inactivate IgM. The DTT-treated serum or plasma is then incubated at 37 °C for 30–60 min to ensure complete IgM inactivation.

The RBCs of blood group A or B are prepared as a 0.8% suspension in normal saline. The patient’s serum is serially diluted two-fold, typically ranging from 1:2 to 1:1024, although the dilution range may be adjusted depending on the expected titer levels. For each dilution, a defined serum volume (e.g., 25 µL) is mixed with an equal or specified volume of RBC suspension (e.g., 50 µL), and the mixture is added to microtubes or wells within the gel or bead card columns.

For IgM antibody detection, the cards are incubated at room temperature (20–25 °C) for 15–30 min before centrifugation. Then, agglutination patterns are visually interpreted according to the manufacturer’s instructions. For IgG antibody detection, the assay is performed using anti-human globulin-coated columns and includes a 37 °C incubation step, typically for 15–30 min, prior to centrifugation and interpretation.

All gel or bead card assays must be centrifuged using a manufacturer-recommended dedicated centrifuge to ensure a consistent and accurate separation of agglutinated and nonagglutinated RBCs within the columns. These centrifuges are specifically designed to accommodate the physical format of CAT cards and to apply the appropriate centrifugal force and orientation necessary for appropriate RBC migration through the matrix. The use of a non-compatible or general-purpose centrifuge might lead to improper RBC settling, resulting in ambiguous agglutination patterns and compromised assay reproducibility. Therefore, the use of a standardized CAT-compatible centrifuge, such as the Bio-Rad ID-Centrifuge, Ortho Workstation II, or equivalent, is essential for result validity and reliability.

The agglutination strength is graded on a scale from 0 (no agglutination) to 4+ (strong agglutination). The antibody titer is defined as the reciprocal of the highest serum dilution yielding a clearly visible agglutination reaction, generally taken as 1+ or stronger. However, some laboratories consider weak positive (w+) reactions as the endpoint, and this inconsistency remains a source of interlaboratory variation. Figure 2 shows an image of the agglutination reaction obtained using CAT.

CAT is generally more qualitative in grading the agglutination reaction strength. In addition, it is less time-consuming, uses smaller volumes of serum and RBCs, and can be used as part of an automated system. In addition, CAT is more effective in standardizing the titer readings because it not only eliminates the inter-reader variability but also produces a stable and reproducible result [16].

However, Tanabe [19] pointed out that the glass bead column agglutination method shows worse reproducibility with a higher maximum titer compared to the gel column agglutination method. Furthermore, the serum-to-cell ratio is slightly higher in the CAT using glass beads. The serum-to-cell ratios suggested by the manufacturer’s instructions differ depending on the column ingredient; hence, the two methods must be distinguished when comparing the target levels of the ABO antibody titer.

Although some reports compared antibody titers measured by CAT and CTM, some studies indicated that the anti-A antibody titers tend to be higher with CAT than with CTM [16,20,21]. However, for the anti-B antibodies, the results are inconsistent across different reports [20,22]. Regardless of whether CAT or CTM is used, the specific criteria for endpoint determination, such as cell concentration, serum treatment, incubation conditions, and agglutination grading, often vary across reports. Given these variations, along with the subjective nature of CTM-based antibody titration, it is difficult to draw consistent conclusions when comparing the results from CTM and CAT.

Given these considerations, CAT offers a promising platform for the semiquantitative evaluation of anti-ABO antibodies; however, it requires rigorous standardization and method transparency to ensure consistency across institutions and clinical applicability, particularly in transplant immunology.

## 7. Flow Cytometry Method (FCM)

FCM has emerged as a promising method for the quantification of anti-A and anti-B antibodies, conferring several advantages over conventional hemagglutination-based assays. One of its key strengths lies in its high reproducibility and objectivity [19]. Unlike CTM and CAT, which rely on the visual assessment of agglutination, FCM provides quantitative results based on fluorescence intensity, thereby reducing interoperator variability. Moreover, FCM allows for isotype-specific detection of immunoglobulins, such as IgG1, IgG2, and IgM, using fluorochrome-conjugated secondary antibodies [23]. This approach facilitates more nuanced immunoprofiling, which can be critical in the context of ABOi KTx, where IgG subclasses might differentially impact the graft outcomes.

An additional advantage of FCM is that it typically does not require prior inactivation of IgM antibodies with reducing agents, such as DTT, as binding can be selectively assessed using isotype-specific antibodies. Consequently, this approach simplifies the workflow and minimizes the potential loss of IgG activity due to DTT treatment.

However, FCM also presents notable limitations. To prevent hemagglutination and allow the accurate detection of cell-bound antibodies, RBCs must be chemically fixed using agents such as formaldehyde and glutaraldehyde. This fixation process introduces variability in epitope preservation and antigen expression levels and can complicate interlaboratory standardization. Furthermore, the method is not yet widely adopted in clinical laboratories, and its use in prospective clinical studies on ABOi transplantation remains limited. Its limited application restricts its clinical validation and the availability of large-scale comparative data with traditional methods.

Despite these challenges, a few studies have demonstrated the utility of FCM in monitoring anti-ABO antibody levels during desensitization or after transplantation [23,24]. For example, Stussi et al. proposed a semiquantitative FCM-based assay capable of separately measuring IgG and IgM anti-A/B antibodies, which correlated well with the clinical response in a limited series of ABOi KTx [24].

Figure 3 shows schematic overview of the FCM. In a standard FCM protocol for ABO antibody titration, group A or group B RBCs, either from a single donor or a pooled sample, are washed and resuspended at a concentration of 0.5% to 1% in PBS. One of the critical preparatory steps is RBC fixation using 0.05% to 0.1% glutaraldehyde for 10–15 min at room temperature. This fixation is essential for preventing spontaneous hemagglutination during serum incubation and preserving consistent surface antigen expression. Following fixation, RBCs are thoroughly washed with PBS and resuspended in PBS containing 0.1% bovine serum albumin to minimize nonspecific binding. The patient serum samples are then serially diluted, typically in two-fold steps ranging from 1:2 to 1:2048, and incubated with equal volumes of fixed RBC suspension at room temperature for approximately 30 min with gentle agitation. The RBCs were then washed thrice to remove unbound serum antibodies. Subsequently, fluorochrome-labeled secondary antibodies specific for human IgG or IgM (e.g., FITC-conjugated anti-IgG or PE-conjugated anti-IgM) are added to the cell suspension and incubated for 30 min at 4 °C in the dark. A final series of washing steps is performed to remove unbound secondary antibodies.

The stained RBCs are then resuspended in buffer and analyzed using a flow cytometer. A minimum of 10,000 RBC events is acquired per sample, and gating is performed using forward and side scatter to exclude debris. The fluorescence intensity of the bound antibodies is quantified as mean or geometric mean fluorescence intensity (MFI). The results are expressed as MFI ratios relative to the appropriate negative controls. Positivity thresholds are often determined using the mean plus three standard deviations of the negative control values or from the receiver operating characteristic curve analysis.

## 8. Enzyme-Linked Immunosorbent Assay (ELISA)-Based Method

ELISA has been explored as an alternative method to traditional hemagglutination-based assays for quantifying anti-A and anti-B antibodies. It offers an objective and highly sensitive detection of antibodies and can be adapted for isotype-specific measurements. In a standard ELISA protocol [25,26], 96-well polystyrene flat-bottom microtiter plates are coated with synthetic or extracted A or B carbohydrate antigens diluted in carbonate buffer (pH 9.6) and incubated overnight at 4 °C. The optimal concentration of synthetic or extracted A or B carbohydrate antigens should be determined in preliminary experiments. Plates are then washed three times with PBS containing 0.05% Tween 20, and non-specific binding sites are blocked with a blocking buffer, such as PBS containing 0.5% Tween 20 and 2% bovine serum albumin (BSA), for 1–2 h at 37 °C. After washing with PBS, appropriately diluted patient serum samples are applied and incubated for 1–2 h at room temperature. Following another PBS wash, plates are incubated with enzyme-conjugated secondary antibodies (anti-human IgM or IgG), and antigen–antibody complexes are visualized using 3,3′,5,5′-tetramethylbenzidine (TMB) liquid substrate. Optical density (OD) is measured with an ELISA reader. Similarly to the flow cytometry-based method, DTT treatment for IgM inactivation is not required, allowing isotype-specific detection.

Despite the abovementioned theoretical advantages, ELISA has not gained widespread clinical adoption for anti-ABO antibody monitoring, particularly in the context of ABOi organ transplantation. The method is technically demanding, requires a standardization of the antigen source and the coating efficiency, and is less well correlated with the hemagglutination titers traditionally used to guide clinical decisions [26]. ELISAs typically employ synthetic A and B carbohydrate antigens instead of native glycoprotein- or glycolipid-bound ABO antigens presented on RBCs; hence, the conformational and spatial presentations of the epitopes might differ significantly. In general, nonspecific binding and cross-reactivity—driven by factors such as Fc-receptor interactions, heterophilic antibodies, rheumatoid factors, or incomplete blocking—are well-documented causes of false-positive results in ELISA. These mechanisms could impact ELISA protocols for anti-A and anti-B antibody detection in a similar manner, particularly when synthetic antigens are used without rigorous blocking and specificity controls.

## 9. ABO Subtype Antibody Assessment (Glycan Microarray and Multiplex Bead Immunoassay)

Recent advances in glycan microarray technology have enabled a detailed analysis of anti-ABO antibodies beyond what traditional hemagglutination assays can provide. ABO blood group antigens exist in multiple glycan subtypes, specifically types I to VI, which differ in their core saccharide structures [27,28,29]. Importantly, while RBCs express a mixture of these subtypes, immunohistochemical studies have shown that vascular endothelial cells—the primary targets in solid organ transplantation—predominantly express the subtype II ABO antigen [30,31]. This difference is clinically relevant because ABMR in ABOi transplantation is most likely driven by antibodies against ABO blood group antigens expressed on endothelial subtype antigens rather than those expressed on RBCs.

To address this immunological specificity, Bentall et al. developed a glycan microarray incorporating synthetic A-subtype-specific glycans, including both trisaccharide and tetrasaccharide forms [31]. The tetrasaccharides particularly allow for a precise subtype discrimination (A-I to A-IV), while the simpler trisaccharide form represents the minimal common epitope. In this platform, patient plasma is incubated on slides printed with these glycan-conjugates. Antibody binding is then detected using fluorescent secondary antibodies specific for IgG or IgM. This method enables an independent quantification of the IgG and IgM responses to each ABO glycan subtype, reported as MFI, which allows for a reproducible and quantitative analysis.

Evaluating the antibody response specifically to subtype II is important in ABOi organ transplantation. When comparing the reactivities of anti-A antibodies by subtype [31], the anti-A-subtype II IgM levels have been shown to exhibit a significant positive correlation with IgM levels against all other A subtypes, suggesting that IgM antibodies exhibit cross-reactivity across multiple A subtypes, which indicates their polyvalent nature. In contrast, anti-A-subtype II IgG have been shown to exhibit a low correlation with the IgG antibodies against other subtypes, depicting a significant correlation only with anti-A-VI IgG. This result implied that IgG antibodies have more specific recognition, and A-subtype II and A-subtype VI might share similar glycotopes. In clinical samples from patients undergoing ABOi KTx, IgG specific to subtype II remains suppressed after immunoadsorption, whereas antibodies to other subtypes often reaccumulate. This pattern contrasts with the therapeutic plasma exchange, which nonselectively reduces all ABO antibodies, regardless of the subtype. These findings highlight the unique ability of immunoadsorption in preferentially depleting clinically relevant ABO glycan subtype II antibodies, which might contribute to its efficacy in desensitization protocols.

The glycan microarray results also show only a moderate correlation with conventional hemagglutination titers, indicating that hemagglutination may not fully capture the biologically significant anti-subtype II responses [31]. Recognizing this limitation, a newer technological adaptation of this glycan array platform has been implemented using the Luminex technology, which is widely adopted in human leukocyte antigen (HLA) antibody testing. Recent studies [32] have demonstrated that this bead-based system allows a multiplexed, high-throughput analysis of the ABO subtype-specific antibodies with enhanced precision and standardization. Importantly, it enables the specific and quantitative measurement of anti-A subtype II antibodies—those most relevant for predicting endothelial injury and rejection in ABOi transplantation.

Nevertheless, these glycan-based assays are not without limitations. The synthetic presentation of glycans on a solid surface may conformationally differ from their native orientation on cell membranes, potentially altering antibody binding. Some glycotopes may be cryptic or exposed only under artificial conditions. Thus, while glycan microarrays offer a powerful and standardized tool to dissect the complexity of anti-ABO responses, their findings must be interpreted in conjunction with functional assays and clinical observations to fully assess the immunological risk of transplantation.

## 10. CD31-ABO Microarray

Isohemagglutinin assays employing RBCs are the most commonly used assays for measuring the antibody titer in ABOi organ transplantation. However, isohemagglutinin antibody titers do not always correlate with the clinical outcome; for instance, an acute ABMR does not occur in some patients with high antibody titers, and vice versa [33,34,35,36]. This finding might be attributed to the differences between the ABO antigens expressed on RBCs and those expressed on vascular endothelial cells, as described above. We identified Pecam1 (CD31) as the most abundant protein linked to the ABO antigens expressed on kidney endothelial cells, which is different from Band 3 or Band 4.5, which is linked to the ABO antigens expressed on RBCs [37]. Based on these findings, we developed a method for measuring antibody titers against the ABO antigens expressed on vascular endothelial cells by immobilizing the ABO glycan antigens expressed on CD31 onto a microarray [7].

To develop a system for elucidating ABO antigens in a vascular endothelial context, recombinant CD31 proteins (rCD31) carrying ABO carbohydrate structures were produced using genetically engineered HEK293 cells (Figure 4). By introducing specific glycosyltransferases, HEK293 cells were modified to express H- (HEK293H), A- (HEK293A), or B-type (HEK293B) glycan antigens. The extracellular domain of human CD31 was cloned and expressed in these cell lines, resulting in FLAG-tagged rCD31 proteins that are specific to each ABO type (H-CD31, A-CD31, and B-CD31). These recombinant proteins were then purified from the culture supernatant through anti-FLAG affinity chromatography and quantified spectrophotometrically. The purified rCD31 proteins (H-CD31, A-CD31, and B-CD31) were immobilized onto epoxysilane-coated glass slides using a noncontact microarray printer. After blocking, the microarrays were incubated with diluted human plasma to detect the anti-ABO antibodies. Separate fluorescent secondary antibodies were used to detect IgG and IgM binding. The fluorescence signals were quantified using a dedicated scanner and analysis software. The antibody levels were calculated by subtracting the H-CD31 signal (serving as a baseline) from the A-CD31 or B-CD31 signal.

The ABO glycans were compared between rCD31 used for the CD31-ABO microarray and CD31 derived from normal human kidney by mass spectrometry analysis, which revealed that the CD31-ABO microarray mimicked the ABO blood group antigens on human kidney endothelial cells. Unlike conventional arrays or ELISA systems that utilize artificially synthesized glycans conjugated to carrier proteins, our approach involved the expression of ABO carbohydrate antigens directly on the CD31 protein backbone. This design mimicked the natural glycosylation patterns found on the vascular endothelial cells in vivo. Consequently, the CD31-based microarray is thought to more accurately reflect a physiologic ABO antigen presentation and offers a more transplant-relevant method for measuring anti-ABO antibody responses compared to previously reported techniques. To validate the clinical efficacy of the CD31-ABO microarray, 252 plasma samples, including volunteers, hemodialysis patients, and transplant recipients, were examined [37]. In transplant recipients, any initial IgG or IgM antibody intensity > 30,000 against the donor blood type in the CD31-ABO microarray showed higher sensitivity, specificity, and positive and negative predictive values of acute ABMR compared to CTM. In A-incompatible kidney transplantation, the sensitivity and specificity of the CD31-ABO microarray for predicting ABMR were 83.3% and 94.0%, respectively, while in B-incompatible transplantation, they were 60.0% and 100%, respectively [37]. A prospective study will be needed to determine whether this cutoff value is associated with the prediction of ABMR and graft survival after ABOi KTx.

In some ABOi KTx cases, patient sera with high antibody titers, as measured by CTM, showed a low reactivity when analyzed using the CD31-ABO microarray [38]. These findings suggest that the CD31-ABO microarray might enable the selective detection of antibodies that specifically recognize the ABO antigens expressed on vascular endothelial cells.

No consensus has yet been reached on which desensitization protocol and dose of immunosuppressive drugs to use—or how far antibody titers should be reduced—before ABOi organ transplantation; hence, an assay that accurately gauges the endothelial risk is highly valuable. The CD31-ABO microarray could fill this gap, as it may allow clinicians to avoid excessive immunosuppression (and subsequent infections) in low-risk recipients while directing a more aggressive therapy toward those who truly remain at immunologic risk.

## 11. Conclusions and Future Directions

Table 1 presents a summary of the antibody measurement methods discussed in this review. Numerous studies have reported anti-ABO antibody titers in the context of ABOi organ transplantation; however, the comparison of titers measured using different methods is of limited value. Even for one particular assay, the absence of standardized protocols and the inherently subjective nature of conventional methods make it difficult to draw meaningful comparisons across institutions or time periods. This issue raises important questions about the clinical relevance of antibody titers measured under inconsistent conditions. In contrast, a meaningful evaluation—similar to that achieved in the field of HLA antibodies—requires the use of standardized platforms that allow for reliable and reproducible comparisons. After years of stagnation in this area, novel technologies, such as glycan subtype and CD31-based microarrays, have emerged. These tools offer endothelial cell-specific detection of anti-ABO antibodies and enable more objective, quantitative analyses. With the adoption of these innovative methods, the assessment of anti-ABO antibody titers is anticipated to become more scientifically rigorous and clinically relevant in the management of ABOi organ transplantation.

## Figures and Tables

**Figure 1 antibodies-14-00078-f001:**
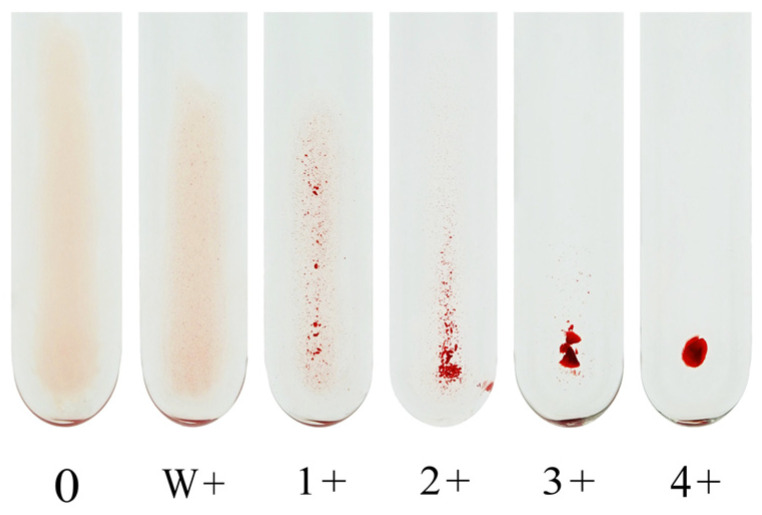
An image of the agglutination reaction obtained using the conventional tube method. W: weak.

**Figure 2 antibodies-14-00078-f002:**
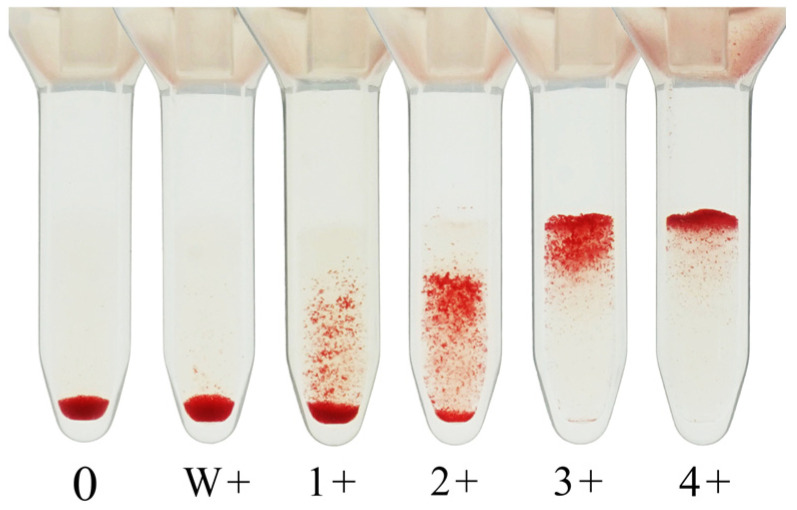
An image of the agglutination reaction obtained using the column agglutination technique. W: weak.

**Figure 3 antibodies-14-00078-f003:**
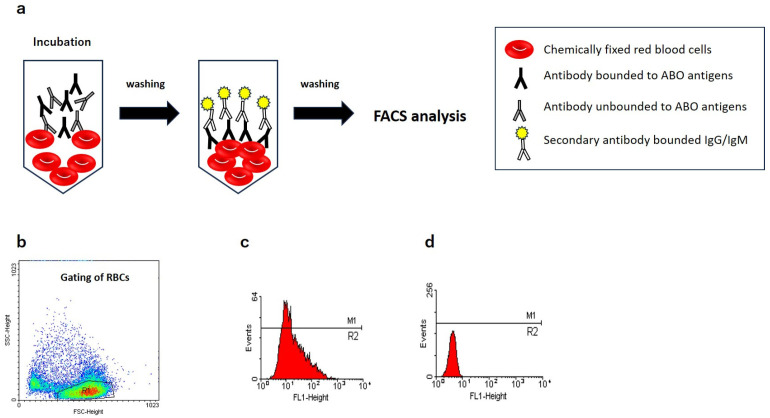
Schematic overview of the FCM for anti-A/B antibodies. (**a**) Blood group A or B RBCs are incubated with patient serum. After washing away unbound antibodies, they are incubated with a secondary antibody. (**b**) Gating RBCs. (**c**) Type B RBCs are incubated with type O serum, which shows a positive reaction. (**d**) Type B RBCs are incubated with type B serum, which shows a negative reaction. RBCs: red blood cells.

**Figure 4 antibodies-14-00078-f004:**
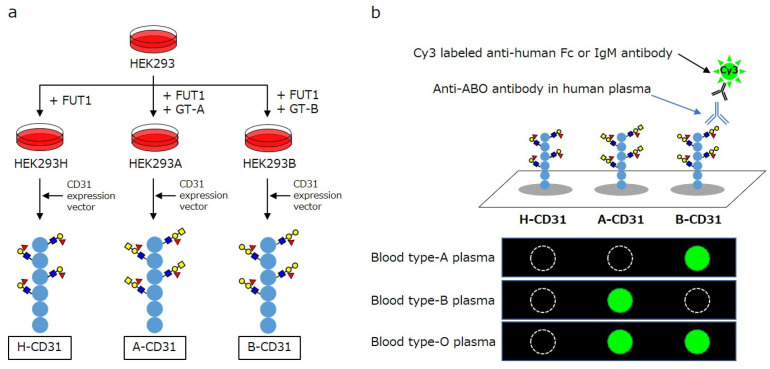
Schema of CD31 linked to ABO carbohydrate antigen microarray [Reproduced from our previous publication [37]. (**a**) The development of recombinant CD31 proteins containing ABO carbohydrate antigen. (**b**) Analyzing anti-A and B antibodies levels in CD31-ABO microarray. HEK: human embryonic kidney, FUT1: α1,2 fucosyltransferase, GT-A: α1,3 *N*-acetylgalactosaminyltransferase, GT-B: α1,3 galactosyltransferase.

**Table 1 antibodies-14-00078-t001:** Summary of antibody assessment in ABO incompatible transplantation.

Method	Need for IgM Removal	Cost	Target Antigen	Objectivity	Advantages	Disadvantages
Conventional Tube Method (CTM)	Yes (recommended)	Low	Native ABO antigens on RBCs	Low (subjective)	Widely used; low cost; simple	Subjective; inter-observer variability; poor reproducibility
Column Agglutination Technique (CAT)	Yes (recommended)	Moderate	Native ABO antigens on RBCs	Moderate	More standardized than CTM	Still partially subjective; limited standardization
Flow Cytometry (FCM)	No	Moderate	ABO antigens on Chemically fixed RBCs	High	Quantitative; sensitive; can detect low-level antibodies	Requires specialized equipment and training; Limited clinical validation
ELISA	No	High	Synthetic ABO oligosaccharides or glycoproteins	High	Quantitative; adaptable	Limited clinical validation
ABO Glycan Microarray	No	High	Synthetic A/B oligosaccharide subtypes (I–VI)	High	Subtype-specific analysis; quantitative	Technically complex; not widely available
CD31-ABO Microarray	No	High	Recombinant CD31 with ABO antigens (endothelial-like)	High	Endothelial-specific ABO detection	Not yet standard; high cost; requires recombinant protein

IgM: immunoglobulin M, RBCs: red blood cells, ELISA: enzyme-linked immunosorbent assay.

## Data Availability

As this manuscript is a review article that synthesizes findings from previously published literature, no new data were generated or analyzed. Accordingly, a specific Data Availability Statement is not applicable.

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
