# Peer review of "Advancing Antibody Titer Assessment in ABO-Incompatible Transplantation"

_2073-4468, 2025, doi:10.3390/antib14030078_

Round 1
Reviewer 1 Report
Comments and Suggestions for Authors
This review paper discusses current and emerging methods for assessing anti-ABO antibody titers in ABO-incompatible (ABOi) transplantation. Beyond the conventional tube method, it describes techniques such as column agglutination, flow cytometry, and enzyme-linked immunosorbent assay, as well as advanced platforms including glycan subtype microarrays and CD31-based endothelial antigen microarrays. These newer approaches offer greater sensitivity, quantitative readouts, and endothelial specificity. The authors emphasize that assays targeting ABO antigens on vascular endothelial cells may more accurately predict antibody-mediated rejection and support standardized titer assessment across institutions, ultimately improving transplant safety and outcomes.
As a transplant immunologist, I found the discussion particularly informative; for example, the section on CD31-based microarrays introduced an endothelial-specific detection strategy that could refine risk assessment beyond traditional RBC-based assays. Notably, this technique was developed by the authors themselves (Reference 43).
I have no concerns about the content of this review.
Author Response
This review paper discusses current and emerging methods for assessing anti-ABO antibody titers in ABO-incompatible (ABOi) transplantation. Beyond the conventional tube method, it describes techniques such as column agglutination, flow cytometry, and enzyme-linked immunosorbent assay, as well as advanced platforms including glycan subtype microarrays and CD31-based endothelial antigen microarrays. These newer approaches offer greater sensitivity, quantitative readouts, and endothelial specificity. The authors emphasize that assays targeting ABO antigens on vascular endothelial cells may more accurately predict antibody-mediated rejection and support standardized titer assessment across institutions, ultimately improving transplant safety and outcomes.
As a transplant immunologist, I found the discussion particularly informative; for example, the section on CD31-based microarrays introduced an endothelial-specific detection strategy that could refine risk assessment beyond traditional RBC-based assays. Notably, this technique was developed by the authors themselves (Reference 43).
I have no concerns about the content of this review.
[Response] Thank you very much for your thoughtful comments.
Reviewer 2 Report
Comments and Suggestions for Authors
This is definitely a relevant topic, although not modern but with significant updates. A very well written manuscript of appropriate length and language with a concise presentation of the ABOi antibody detection methods we utilise currently in kidney transplantation as well as novel approaches and their benefits/drawbacks. Serves as a thorough review on the topic I would recommend to a person unfamiliar with the topic to go through before proceeding deeper in the field. The summarising table is very useful.
I would recommend adding a couple of more pictures of possible.
Author Response
This is definitely a relevant topic, although not modern but with significant updates. A very well written manuscript of appropriate length and language with a concise presentation of the ABOi antibody detection methods we utilise currently in kidney transplantation as well as novel approaches and their benefits/drawbacks. Serves as a thorough review on the topic I would recommend to a person unfamiliar with the topic to go through before proceeding deeper in the field. The summarising table is very useful.
I would recommend adding a couple of more pictures of possible.
[Response] We appreciate the comment. We added figure 3 in the main text.
Reviewer 3 Report
Comments and Suggestions for Authors
In this Review Paper Tasaki et al. Explore the topic of “Advancing Antibody Titer Assessment in ABO-Incompatible Transplantation". The Title is well-crafted and effectively draw the reader’s attention. The manuscript provides a valuable overview of current practices and emerging approaches in this area. Below are several specific comments that may help enhance the clarity, depth, and overall quality of the paper.
- What are the existing techniques of measuring anti-ABO antibodies, and how do their sensitivity, specificity, and reproducibility compare when used in ABO-incompatible kidney transplantation (ABOi KTx)?
- What do the primary issues and limitations related to traditional antibody titration methods include and how do they affect the standardization and clinical interpretation of anti-ABO antibody levels?
- What is the impact of IgM inactivation on the accuracy of IgG anti-ABO antibody assay and what are the most dependable methods of selective IgM inactivation in clinical practice?
- given the high inter institutional variability associated with CTM and CAT, what are the authors suggestions or intentions to establish a standard reference or a system of calibration that might enable cross-platform and cross-laboratory comparison of anti-ABO antibody titers?
- With respect to the CD31-ABO microarray, the authors might specify how the cutoff threshold of >30,000 fluorescence intensity was calculated, and whether this threshold has been prospectively determined with respect to clinical outcome such as acute ABMR or graft survival.
- What are the guidelines that the authors can give relating to the interpretation of differences in the event of such differences in clinical settings (e.g., CTM vs. CD31-microarray), and how do these findings need to be prioritized or combined by clinicians to inform decision-making in transplantation of ABO-incompatible organs?
Author Response
In this Review Paper Tasaki et al. Explore the topic of “Advancing Antibody Titer Assessment in ABO-Incompatible Transplantation". The Title is well-crafted and effectively draw the reader’s attention. The manuscript provides a valuable overview of current practices and emerging approaches in this area. Below are several specific comments that may help enhance the clarity, depth, and overall quality of the paper.
[Comment 1] What are the existing techniques of measuring anti-ABO antibodies, and how do their sensitivity, specificity, and reproducibility compare when used in ABO-incompatible kidney transplantation (ABOi KTx)?
[Response 1] We appreciate your comment. We summarize all currently used and emerging assays in the manuscript—CTM, CAT (gel and glass-bead card), flow cytometry, ELISA, glycan microarrays (ABO subtypes I–VI), and the endothelial-specific CD31-ABO microarray. Unfortunately, there are no reports directly comparing the sensitivity, specificity, and reproducibility of each method in predicting antibody-mediated rejection (ABMR) after ABO-incompatible organ transplantation. However, for the CD31-ABO array, there are reports demonstrating its sensitivity and specificity for predicting ABMR compared with CTM, and we have added these results to the manuscript (P14, L581). In addition, there are reports comparing antibody titers measured by different methods (Ref. 26, 27), but since the detailed procedures for antibody measurement vary among institutions, we believe that these studies cannot be meaningfully applied to clinical practice at other centers.
[Comment 2] What do the primary issues and limitations related to traditional antibody titration methods include and how do they affect the standardization and clinical interpretation of anti-ABO antibody levels?
[Response 2] We appreciate your comment. We explicitly detail the major issues: lack of harmonized protocols (reagent RBC choice, serum-to-cell ratio, incubation time/temperature), endpoint definition (w+ vs 1+), operator subjectivity, and inconsistent IgM inactivation—each of which drives inter-observer and inter-laboratory variability and complicates clinical interpretation. The details are provided in the main text. Ideally, by establishing a rigorous protocol as a global standard and achieving harmonization, it would become possible to compare the interpretation of antibody titers across institutions. However, there remains the limitation that such assessment is inherently objective in nature. Please see the details written in the section of Conventional Tube Method (P6, L228 and P7, L277).
[Comment 3] What is the impact of IgM inactivation on the accuracy of IgG anti-ABO antibody assay and what are the most dependable methods of selective IgM inactivation in clinical practice?
[Response 3] We appreciate your comment. We explain that residual IgM can cause false-positive/overestimated IgG titers and therefore bias risk assessment. Among available approaches (DTT, 2-ME, heat, enzymatic digestion, immunoaffinity), DTT is presented as the most reliable and widely adopted for selective IgM inactivation with preservation of IgG function. We also clarify that many laboratories still perform “IgG” titration without DTT; in such cases, results should be reported as “total antibodies” rather than true IgG, per published guidance. We described the details in the manuscript (P4, L162)
[Comment 4] given the high inter institutional variability associated with CTM and CAT, what are the authors suggestions or intentions to establish a standard reference or a system of calibration that might enable cross-platform and cross-laboratory comparison of anti-ABO antibody titers?
[Response 4] Our review highlights existing harmonization efforts (e.g., CAP uniform procedure; annual external quality assessment in Japan) and practical steps (use of dedicated CAT-compatible centrifuges; adopting a 1+ endpoint) to reduce variability. If a global standard protocol could be established and implemented, inter-institutional variability could be reduced compared with the current situation. We emphasize that truly comparable, cross-platform/cross-laboratory assessment will be best enabled by standardized, objective platforms—specifically glycan-subtype and CD31-based microarrays—which provide instrument-read quantitative signals tied to endothelial biology.
[Comment 5] With respect to the CD31-ABO microarray, the authors might specify how the cutoff threshold of >30,000 fluorescence intensity was calculated, and whether this threshold has been prospectively determined with respect to clinical outcome such as acute ABMR or graft survival.
[Response 5] The diagnostic potential of the CD31-ABO microarray was determined by calculating the receiver operating characteristic (ROC) curve plotted to evaluate the sensitivity and specificity for predicting acute ABMR after ABOi KTx. Any initial IgG or IgM Ab levels against donor blood type >30,000 in the CD31-ABO microarray showed high sensitivity, specificity, positive predictive value (PPV), and negative predictive value (NPV) in both anti-A and anti-B Abs. This is described in Ref. 43. We added the sentence that “A prospective study will be needed to determine whether this cutoff value is associated with the prediction of ABMR and graft survival after ABO-incompatible kidney transplantation” (P14, L584).
[Comment 6] What are the guidelines that the authors can give relating to the interpretation of differences in the event of such differences in clinical settings (e.g., CTM vs. CD31-microarray), and how do these findings need to be prioritized or combined by clinicians to inform decision-making in transplantation of ABO-incompatible organs?
[Response 6] We believe that measurement methods based on RBCs are not accurate for evaluating the response to ABO antigens in ABOi organ transplantation. Therefore, rather than comparing conventional methods such as CTM or CAT with approaches that assess ABO antigen reactions on vascular endothelial cells, we consider it more important to prioritize the results of antibody measurements that are specific to vascular endothelial cells.